# Vanillic Acid Ameliorates Demyelination in a Cuprizone-Induced Multiple Sclerosis Rat Model: Possible Underlying Mechanisms

**DOI:** 10.3390/brainsci14010012

**Published:** 2023-12-22

**Authors:** Sally M. Safwat, Mahmoud El Tohamy, Moutasem Salih Aboonq, Amaal Alrehaili, Ahmad A. Assinnari, Abdulrahman S. Bahashwan, Ahmed A. ElGendy, Abdelaziz M. Hussein

**Affiliations:** 1Department of Medical Physiology, Faculty of Medicine, Mansoura University, Mansoura 35516, Egypt; sallysafwat186@gmail.com (S.M.S.); dr.m.eltohamy@gmail.com (M.E.T.); dr_ahmedelgendy@hotmail.com (A.A.E.); 2Department of Medical Physiology, College of Medicine, Taibah University, KSA, Madinah 42353, Saudi Arabia; aboonq@yahoo.co.uk (M.S.A.); afrehaili@taibahu.edu.sa (A.A.); aassinnari@taibahu.edu.sa (A.A.A.); asbahashwan@taibahu.edu.sa (A.S.B.)

**Keywords:** cuprizone, demyelinating, vanillic acid, Nrf2, neuroinflammation

## Abstract

Objective: To investigate the effect of vanillic acid (VA) on a Cuprizone (Cup) demyelinating rat model and the mechanisms behind such effect. Methods: Thirty adult male Sprague Dawley (SD) rats were randomly divided into three groups: control, Cuprizone, and VA groups. Cuprizone was administrated at a dose of 450 mg/kg per day orally via gastric gavage for 5 weeks. The nerve conduction velocity (NCV) was studied in an isolated sciatic nerve, and then the sciatic nerve was isolated for histopathological examination, electron microscope examination, immunohistochemical staining, and biochemical and PCR assay. The level of IL17 was detected using ELISA, while the antioxidant genes Nrf2, HO-1 expression at the level of mRNA, expression of the myelin basic protein (MBP), interferon-gamma factor (INF)-γ and tumor necrosis factor (TNF)-α, and apoptotic marker (caspase-3) were measured using immunohistochemistry in the sciatic nerve. Results: There was a significant reduction in NCV in Cup compared to normal rats (*p* < 0.001), which was markedly improved in the VA group (*p* < 0.001). EM and histopathological examination revealed significant demyelination and deterioration of the sciatic nerve fibers with significant improvement in the VA group. The level of IL17 as well as the expression of INF-γ and caspase-3 were significantly increased with a significant reduction in the expression of MBP, Nrf2, and HO-1 in the sciatic nerve (*p* < 0.01), and VA treatment significantly improved the studied parameters (*p* < 0.01). Conclusion: The current study demonstrated a neuroprotective effect for VA against the Cup-induced demyelinating rat model. This effect might be precipitated by the inhibition of inflammation, oxidative stress, and apoptosis.

## 1. Introduction 

Multiple sclerosis (MS) is a chronic autoimmune disorder of the nervous system, characterized by progressive demyelination and inflammation [1]. There are around 2.5 million MS patients globally. Clinically, MS has been classified into relapsing–remitting MS (RRMS), primary progressive MS (PPMS) (active and not active), and secondary progressive MS (SPMS) (active and not active) [2]. MS’s exact cause is still unknown; however, environmental or genetic factors may play a role. Multiple foci of inflammatory reactions, microgliosis, astrocytosis, oligodendrocyte depletion, demyelination, and axonal degeneration are all part of the pathological process of MS [3]. A better understanding of its underlying mechanisms is mandatory for the discovery of new lines for its management. MS can be replicated in experimental animals using different models including toxin-induced demyelination (Cuprizone or Cup), virus-induced demyelination, and experimental autoimmune encephalitis (EAE) which were established for the proper understanding of its pathophysiological mechanisms [4]. Cuprizone is a frequently used and well-liked paradigm in which Cuprizone-fed animals experience oligodendrocyte cell loss that, when interrupted, results in spontaneous remyelination that mimics the relapsing–remitting stage of multiple sclerosis. It acts as a major copper chelating agent that causes oligodendrocytes to undergo apoptosis in particular, which results in continuous demyelination [5].

Interleukin-17A (IL-17A commonly known as IL-17) has a pivotal role in the pathogenesis of MS [6]. Th-17 cells together with oligodendrocytes and astrocytes in the CNS are responsible for the production of IL-17 [7]. Upon binding to its receptor, IL-17 activates a variety of signaling pathways, including nuclear factor-kappa B (NF-kB) [8]. Additionally, IL-17 induces the synthesis of chemokines and the influx of neutrophils in microglia and astrocytes, which contribute to the onset or progression of MS [9]. Additionally, prior research revealed that the concentrations of TNF-α and IFN-γ are notably elevated in MS pathogenesis and linked to the disease’s progression [10]. Additionally, prior investigations have shown that matrix metalloproteinases (MMPs), particularly MMP-9, have a role in the pathogenesis of MS [11]. The myelin basic protein (MBP), which is regarded as one of the fundamental components of a myelin sheath, is destroyed as a result of the digestive action of MMP-9 [12]. 

Natural substances are commonly used for treating various disorders. Conjugated acid of vanillate, often known as vanillic acid (VA), is a naturally occurring phenolic acid. It is made from the Angelica sinensis plant, which is utilized in conventional Chinese medicine. Vanillin is a common flavoring ingredient used in the food sector, cosmetics, and medicines. It has anti-inflammatory, hepatoprotective, cardioprotective, and antioxidant properties [13]. The therapeutic potential of vanillic acid in multiple neurological disorders is evident in previous studies and indicates its antioxidant, anti-inflammatory, and anti-apoptotic effects [14,15,16]. In the current study, we hypothesized that vanillic acid could be a promising and accessible neuroprotective agent in the management of demyelinating diseases such as MS. So, this study aimed to explore the potential protective role of vanillic acid on peripheral nerves’ structure and function in a rat model of MS and the possible underlying mechanisms behind such effect including its antioxidant, antiapoptotic, and anti-inflammatory effect. 

## 2. Materials & Methods

### 2.1. Experimental Animals

Twenty-four male albino rats aged 12–16 weeks old, weighing 150–200 g were bred in the Medical Experimental Research Center (MERC) animal house at Mansoura Faculty of Medicine. Rats were housed in conditions of controlled illumination (12 h light/dark cycle), temperature (20–22 °C), humidity (40–70%), and free access to water and food. All experimental procedures were performed according to the guidelines provided by our local committee of animal research ethics. The research protocol was approved by the Mansoura Medical Research Ethics Committee (IRB) Code Number: R.22.06.1729) on 2 July 2022.

### 2.2. Experimental Design

The study involved 3 experimental groups each containing 8 rats: Group I (control) normal rats were maintained on standard rat chow with 1.5 mL of carboxymethylcellulose (CMC) daily by oral gavage for 5 weeks, Group II (Cuprizone) rats received 450 mg/Kg of Cuprizone dissolved in 1.5 mL of 1% CMC, daily orally for 5 weeks [5], and Group III (Cuprizone treated by vanillic acid) rats received 450 mg/Kg of Cuprizone dissolved in 1.5 mL of 1% CMC, daily orally and 50 mg/kg/day of vanillic acid orally for 5 weeks [17]. Male rats were chosen in the current work based on a previous study by Ünsal and Özcan, [18] who found that both male and females are affected by Cuprizone-induced neurotoxicity but males are affected more than females. At the end of the experimental protocol, rats were humanely sacrificed using an IP injection of pentobarbital overdose of 30 mg/kg for further assessment.

### 2.3. Nerve Conduction Velocity

Each rat’s sciatic nerve was carefully detached from one hind limb using a glass hook, and 5–6 cm of the nerve was excised. The nerve specimens were transferred to a nerve chamber which was then stimulated using a power lab stimulator to capture compound action potentials (CAP). Fresh Krebs solution (119 mM NaCl, 20 mM 10 mM glucose, NaHCO_3_, 4.8 mM KCI, 1.8 mM CaCl_2_, 1.2 mM MgSO_4_, and 1.2 mM KHPO_4_, gassed with a combination of 95% O_2_ and 5% CO_2_) was poured into the nerve chamber. A 37 °C ambient temperature and a pH of 7.4 were maintained [19]. By dividing the distance between electrodes (in meters) by the time interval (in seconds), the conduction velocity (in m/sec) was calculated.

### 2.4. Histological Examination

Nerves were taken out for histological analysis after performing calculations. Samples of the sciatic nerve were preserved in a neutral formalin solution at 10%. Tissues that had been fixed in paraffin were serially cut into 5 µm slices. To assess the degree of neuronal injury and the state of myelination, sciatic nerve slices were stained with hematoxylin & eosin and Luxol fast blue (LFB) [20].

### 2.5. Electron Microscopy Examination (TEM)

Freshly cut sciatic nerve slices from each group were fixed in 4% glutaraldehyde, following which alterations in ultrastructure were processed and analyzed [21]. The morphological changes in the sciatic nerve were described according to a previous study by Love, S. [22], who reported the EM changes in a rat model of Cuprizone-induced neurotoxicity. Also, the thickness of the myelin sheath in a number of large-diameter fibers was calculated.

### 2.6. Evaluation of IL-17

Sciatic nerve samples were homogenized for the measurement of IL-17 by employing the ELISA technique using ELISA kits (My Biosource, San Diego, CA, USA. Catalog # MBS175940).

### 2.7. Immunohistopathological Examination

Deparaffinization, rehydration, washing, submersion in 3% hydrogen peroxide, pepsin digestion, and antigen retrieval were all performed on the sciatic nerve sections. Sections were then cleaned with phosphate buffer. Each slide was microwave-heated in 10 mmol/L citrate buffer, pH 6.0, for 10–20 min after being incubated for 30 min with 0.3% hydrogen peroxide in methanol. After unspecific binding by serum blocking, the sections were incubated with primary antibodies of interferon-gamma (Cusabio, Wuhan, China. Catalog# CSB-E04578m, dilution 1:100 for 2 h); myelin basic protein (Catalog # PA5-78397, Thermo Fisher Scientific, Waltham, CA, USA for MBP); tumor necrosis alpha (Catalog# MBS438411, My Biosource, San Diego, CA, USA: dilution 1:100 for 2 h); and Caspase-3 (Catalog # PA5-77887, Thermo Fisher Scientific, Waltham, CA, USA, dilution 1:50 for 2 h). The number of positive cells and region of interest (ROI) in 5 high power fields (HPF) were calculated using imageJ software (1.54h, NIH, Bethesda, MD, USA).

### 2.8. PCR Assay of Antioxidant Genes (Nrf2/HO-1)

The expression of antioxidant genes, nuclear erythroid-related factor 2 (Nrf2), and heme oxygenase-1 (HO-1) in sciatic nerve tissue was detected using real-time PCR. The total RNA from the samples was extracted following the manufacturer’s instructions (TRIzol™ Reagent (cat no 15596026, Invitrogen, Waltham, MA, USA). RNA was measured spectrophotometrically, and agarose gel electrophoresis and ethidium bromide staining were used to assess its quality. For cDNA synthesis, a cDNA reverse transcription kit with a high 9 capacity (cat no 4374966, Thermo Fisher Scientific, Waltham, CA, USA) and green master SYBR mix (cat no. k0221, Thermo Fisher Scientific, Waltham, CA, USA) were used. Briefly, 10 μL of RNA sample was mixed with 10 μL of reverse transcription master mix, which also included 3.2 μL of nuclease-free H_2_O, 1 μL of MultiScribe ™ Reverse Transcriptase, 1 μL of RNase Inhibitor, 2 μL of RT Random Primers, and 0.8 μL of dNTP mix (100 mM). After 10 min at room temperature, the mixture was incubated for 120 min at 37 degrees Celsius. Ultimately, cDNA was kept in storage at −80 °C. The sequence of tested genes was as follows: Nrf2 gene [5′ATTGCTGTCCATCTCTGTCAG-3′.(sense) 5′-GCTATTTTCCATTCCCGAGTTAC-3′ (antisense)], HO-1 [5′TGCTTGTTTCGCTCTATCTCC-3′.(sense) 5′-CTTTCAGAAGGGTCAGGTGTC-3′, (antisense)]. Using a thermal cycler, amplification and detection were carried out. Denaturation was performed at 95 °C for 10 min followed by 40 cycles of denaturation at 95 °C for 15 s, annealing at 60 °C for 1 min, and extension at 72 °C for 1 min. The following formula was used to calculate the amount of cDNA: 2-ct (2—[(Ct of the target gene-Ct of GAPDH in treated rats)—(Ct of the target gene—Ct of GAPDH in control rats)]) [23].

### 2.9. Statistical Analysis

The data are expressed as Mean ± SD. One-way ANOVA followed by Tukey^’^s post hoc test were used for measuring the statistical significance among all groups. *p* < 0.05 was considered significant.

## 3. Results

### 3.1. Effect of Vanillic Acid on Cuprizone-Induced Neurophysiologic Changes

To assess the function of the sciatic nerve, the nerve conduction velocity was measured by using PowerLab. The NCV in isolated sciatic nerve in the Cuprizone group was decreased significantly when compared to the control group (*p* < 0.001). On the other hand, it was significantly elevated with the VA group when compared to the Cuprizone group (*p* < 0.001). Moreover, there was no significant change between the vanillic acid-treated group and the normal control group (Figure 1A). Figure 1B–D show a record of the nerve conduction study from the control group, Cuprizone group, and VA group, respectively.

### 3.2. Effect of Vanillic Acid on the Level of Inflammatory Cytokine (IL17) and Antioxidant Gene (Nrf2 and HO-1) Expression at the mRNA Level in Sciatic Nerve Tissues

The level of IL17 in the sciatic nerve showed a significant increase in the Cuprizone group compared to the normal group (*p* < 0.001). On the other hand, it was significantly attenuated in the vanillic acid-treated group when compared with the Cuprizone group (*p* < 0.01) (Figure 2A). 

We also assessed the expression of the antioxidant genes, nuclear Nrf-2 (Nrf-2) and HO-1 at the mRNA level using RT-PCR. One Cuprizone group revealed a significant reduction in the expression of Nrf-2 and HO-1 in the sciatic nerve (*p* < 0.05) compared with the normal group at the mRNA level. In contrast, the vanillic acid-treated group showed significant elevation in Nrf-2 and HO-1 (*p* < 0.001) in comparison with the Cuprizone group. Also, there was a significant elevation in both markers in the vanillic acid-treated group when compared with the normal control group (*p* < 0.01) (Figure 2B,C).

### 3.3. The Effect of Vanillic Acid on Cuprizone-Induced Sciatic Nerve Demyelination

The evaluation of the myelin status was performed through Luxol fast blue (LFB) staining and analysis of MBP via immunostaining of the sciatic nerve. The Cuprizone group exhibited significant demyelination which was proved by a profound reduction in LFB staining intensity (*p* < 0.001) compared with the control group. On the other hand, the vanillic acid-treated group revealed a significant elevation in the intensity of LFB staining (*p* < 0.001) (Figure 3A). Figure 3B–D show representative photomicrographs of LFB staining from the normal control group, Cuprizone group, and vanillic acid-treated group, respectively.

Additionally, the Cuprizone group showed a notable decrease in MBP expression (*p* < 0.05) compared to the normal group. The vanillic acid-treated group revealed significant elevation in MBP expression in the sciatic nerve in comparison with the Cuprizone group (*p* < 0.001) (Figure 4A). Figure 4B–D show representative photomicrographs of MBP immunostaining from the normal control group, Cuprizone group, and vanillic acid-treated group, respectively.

### 3.4. Effect of Vanillic Acid on the Cuprizone-Induced Apoptotic Marker (Caspase-3)

The expression of the caspase-3 protein in the sciatic nerve tissue was determined by immunostaining to provide information about the activation of the apoptotic pathway. Figure 4 shows that the expression of the caspase-3 protein was significantly higher in the Cuprizone group (*p* < 0.001) in comparison to that of the control group. Upon treatment by vanillic acid, its level was significantly decreased (*p* < 0.001) when compared to the Cuprizone group (Figure 5A). Figure 5B–D show representative photomicrographs of caspase-3 immunostaining from the normal control group, Cuprizone group, and vanillic acid-treated group, respectively.

### 3.5. Effect of Vanillic Acid on Cuprizone-Induced Neuroinflammatory Cytokines (IFN-γ and TNF-α)

To assess neuro-inflammation, the expression of IFN-γ and TNF-α at the protein level was analyzed via immunostaining of the sciatic nerve. The Cuprizone group showed a significant increase in IFN-γ (*p* < 0.001) compared to the normal group. In contrast, vanillic acid treatment had markedly decreased IFN-γ (*p* < 0.001) compared to the Cuprizone group (Figure 6A). Figure 6B–D show representative photomicrographs of IFN-γ immunostaining from the normal control group, Cuprizone group, and vanillic acid-treated group, respectively. However, there was a non-significant change in TNF-α among the different studied groups (Figure 7A). Figure 7B–D show representative photomicrographs of TNF-α immunostaining from the normal control group, Cuprizone group, and vanillic acid-treated group, respectively.

### 3.6. Effect of Vanillic Acid on Sciatic Nerve Morphology

The sciatic nerve obtained from normal rats showed normal histological architecture of the nerve fibers which were myelinated, and the myelin sheaths were dense, uniform, and arranged in concentric rings (Figure 8A). In contrast, the sciatic nerve of the Cuprizone group showed that the myelinated fibers were loose, broad, and partially demyelinated with an irregular pattern of clear vacuoles, i.e., bubbling and inflammatory cell infiltrates (Figure 8B). The vanillic acid-treated group revealed regular myelin sheaths similar to rats of the normal control group (Figure 8C).

We also used transmission electron microscopy to examine the ultrastructure of the sciatic nerve and myelin sheath. The thickness of the myelin sheath of large-diameter nerve fibers revealed a significant reduction in the Cuprizone group in comparison to normal control group (*p* < 0.001). The vanillic acid-treated group revealed a significant increase in myelin thickness in comparison to the Cuprizone group (*p* < 0.001) (Figure 9A). The sciatic nerve in normal rats showed large myelinated nerve fibers with an intact and thick myelin sheath (M) and its axonal (A) cytoplasm containing mitochondria (m) with small-diameter nerve fiber axons surrounded by Schwann cells ensheathing each axon in a pocket of its cytoplasm, forming a Remak bundle and normal thickness of endoneurium, and implying normal spacing of nerve fibers within the sciatic nerve (Figure 9B). On the other hand, the Cuprizone group showed endoneurial edema (manifested by distinct displacement between nerve fibers), Wallerian degeneration (W) in axons of large-diameter fibers, and degeneration of non-myelinated nerve fibers surrounded by Schwan cells (Figure 9C). The vanillic acid-treated group exhibited a preserved sciatic nerve structure with intact large- and small-diameter axons near the normal control group (Figure 9D).

## 4. Discussion

The current study was undertaken to analyze the potential beneficial effects of vanillic acid on Cuprizone-induced demyelination of the sciatic nerve in rats. MS is known as a progressive disease of the central nervous system, which might lead to central axon demyelination and subsequent neurological deficits resulting in peripheral or central symptoms. Even though the basic target of MS is the axons of the CNS, there is growing concern about its impact on the PNS. In addition, neuro-pathological as well as electro-physiological examinations point towards the greatest effect of MS on the PNS [24].

Myelin and axonal degeneration and subsequent regeneration and other important pathologies related to MS can be better assessed by the Cuprizone model. Cuprizone administration for 5–6 weeks is enough for consistent demyelination to be achieved when Cup is continued for more than 10 weeks, which will lead to chronic demyelination. Spontaneous remyelination occurs once Cup administration is stopped [5], so to assess the effect of vanillic acid on Cuprizone-induced demyelination we gave it concomitantly with Cuprizone from the start of our experimental period for 5 weeks. To support the occurrence of a peripheral demyelinating process, we investigated the alterations in the structure of the sciatic nerve using H&E, LFB, and MBP staining and transmission electron microscopy. Our study revealed that Cuprizone decreased the LFB staining intensity and MBP expression compared with the control group, suggesting the development of the demyelination of the PNS. Moreover, an electron microscopic examination revealed significant deterioration of the sciatic nerve. Our result was in alignment with that of Yu et al. [25] who stated that Cuprizone induced a significant decrease in the level of MBP in the cerebral cortex and hippocampus together with neuronal pyknosis, degeneration, and demyelination. Ohgomori & Jinno [26] also stated that Cuprizone administration for five weeks induced hippocampus demyelination. Moreover, the ultramicroscopic changes shown by EM in the sciatic nerve in the current study in the Cup group are in agreement with those described by Love [22]. Alternatively, VA treatment markedly attenuated the process of demyelination in the sciatic nerve and improved its microscopic structure and nerve conduction velocity suggesting a neuroprotective effect for VA against Cuprizone-induced neurotoxicity in peripheral nerves. In line with our results, Pawar et al. [27] reported significant improvement in peripheral nerve histology after chronic constriction injury, and Huang et al. [28] reported that VA reversed the histopathological changes of diabetic neuropathy.

Myelin’s main function is to provide a protective coating for the nerve fibers, which enables fast-action potential conduction in the nervous system, providing salutatory conduction of neural impulses at high conduction velocity. In the PNS, myelin is formed mainly by Schwann cells; however, in the CNS, oligodendrocytes are the myelin-producing cells [29]. In the current study, we found demyelination in the sciatic nerve using LFB staining as well as downregulation of MBP in the sciatic nerve of the Cuprizone group. The process of demyelination of nerve fibers is reflected in the conduction velocity of the nerve fibers, which is markedly reduced with demyelination. That is why we examined the conduction velocity in the current study on isolated sciatic nerve fibers. In the current research, we found a significant reduction in conduction velocity in the Cup group compared to the control group, suggesting that demyelination of nerve fibers resulted in a reduction in CV in the sciatic nerve. This finding is in agreement with that of Ünsal, C., & Özcan, M. [18], who reported a significant reduction in sciatic nerve conduction velocity with Cup treatment in both male and female rats. They also reported that males are more sensitive to Cup than females, which is why we chose male rats in the current study. On the other hand, treatment with VA significantly attenuated the process of demyelination in the sciatic nerve, upregulation and expression of MBP, and enhanced the nerve conduction velocity, suggesting a remyelinating effect of VA against Cuprizone-induced neurotoxicity in peripheral nerves. These findings are in line with those reported by Pawar et al. [27] who stated that VA significantly improved the nerve conduction velocity, electrophysiological changes, and nerve histology in a rat model of a neuropathic model. Moreover, Siddiqui et al. [30] demonstrated that vanillic acid mediated an increase in MBP expression, which might be associated with the increased availability of neurotrophins, and through its anti-inflammatory activity, inhibits matrix metalloproteinases (MMPs) such as MMP-9, which is responsible for MBP destruction.

The second aim of the current study was to investigate the mechanisms underlying Cuprizone demyelinating rats. Cuprizone is a copper-chelator which causes mature oligodendrocytes’ destruction. This effect of CUP is attributed to mitochondrial activity inhibiting the copper-dependent monoamine oxidase and cytochrome oxidase enzymes, causing mitochondrial dysfunction either through clustering or enlargement; this is followed by their apoptosis, resulting in their death. Reactive astrocytes and microglia populate in the areas of demyelination and produce pro-inflammatory cytokines (TNF-α, Il-1β and Inf-γ). So, neuroinflammation, oxidative stress, and apoptosis are the main contributors that govern the Cuprizone-induced demyelination process [31].

The first mechanism that we investigated was the process of neuroinflammation in Cuprizone-induced neurotoxicity by measurement of inflammatory cytokines in the sciatic nerve, including IL17, TNFα, and INFγ. Our results revealed that Cuprizone caused a significant increase in inflammatory mediators such as IL-17 and IFN-γ without changes in the expression of TNF-α, and these findings are in line with those of Abdel-Maged et al. [2]. These findings suggest the development of a neuroinflammatory process in the sciatic nerve with Cuprizone feeding. Recently, ELBini and Neili [32] demonstrated that the use of a K channel blocker stimulated the process of the myelination of nerve fibers in a model of multiple sclerosis when introduced early during disease development or during remission and exacerbated the demyelination when introduced in a toxic inflammatory environment during the inflammatory processes. Moreover, vanillic acid treatment significantly decreased these inflammatory markers (IL-17, INF-γ) in the sciatic nerve, which is in parallel with the results of Katary & Salahuddin, [33] that vanillic acid downregulates inflammatory cytokines induced by indomethacin in the gastric mucosa. These findings suggest anti-inflammatory actions of VA in the Cuprizone demyelinating rat model. This anti-inflammatory action of VA was reported by a previous study by Calixto-Campos et al. [34]. Moreover, in the current study, we found that VA therapy caused significant attenuation in INF-γ more than in the control group; however, we do not have any explanation for this finding.

In our research, Cuprizone was also found to increase the expression of proapoptotic marker, caspase-3 in comparison with the control group. This was in agreement with Ghobadi et al. [35] that Cuprizone could initiate apoptosis in oligodendrocytes and this reaction activates microglia/macrophages and leads to the destruction of myelin sheets. Rats treated with vanillic acid exhibited a significant reduction in caspase-3 expression when compared to the Cuprizone group, which was in line with observations from other research groups [36].

Finally, there is a growing belief about the oxidative stress role in multiple sclerosis. Cuprizone administration raises ROS levels inside the oligodendrocytes which trigger or stimulate its apoptosis [37]. Nrf-2 is a redox transcription factor which acts as a main regulator for HO-1 induction in the CNS with the activation of different antioxidants [38]. In the current study, the expression of nuclear Nrf-2 and HO-1 was significantly decreased in the Cuprizone group when compared to the normal group and this was previously reported by Abdel-Maged et al. [2]. However, vanillic acid treatment reversed the reduction of nuclear Nrf-2 and HO-1, and this was in accordance with the results of El-Hefnawy et al. [39] who reported that vanillic acid showed anti-epileptic and neuroprotective effects against PTZ-induced epilepsy which probably might be due to its antioxidant properties and upregulation of the Nrf2/HO-1 pathway.

## 5. Conclusions 

In conclusion, VA might ameliorate the demyelinating process in peripheral nerves associated with Cuprizone-induced neurotoxicity and improve the conduction velocity in peripheral nerves in this rat model. This might be due to its probable upregulation of endogenous antioxidant genes and downregulation of the inflammatory process and apoptotic processes.

## Figures and Tables

**Figure 1 brainsci-14-00012-f001:**
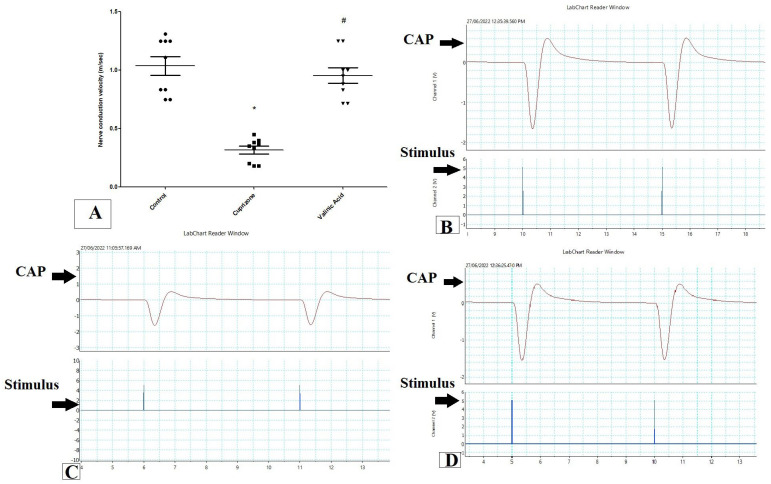
Nerve conduction study in isolated sciatic nerve. (**A**) nerve conduction velocity (m/s) from different groups. Traces of NCS records from the control group (**B**), Cuprizone group (**C**), and VA group (**D**). * significant vs. control group, # significant vs. Cuprizone group. CAP = compound nerve action potential. *p* < 0.05. Red line is the curve of M wave and blue line indicate the stimulus.

**Figure 2 brainsci-14-00012-f002:**
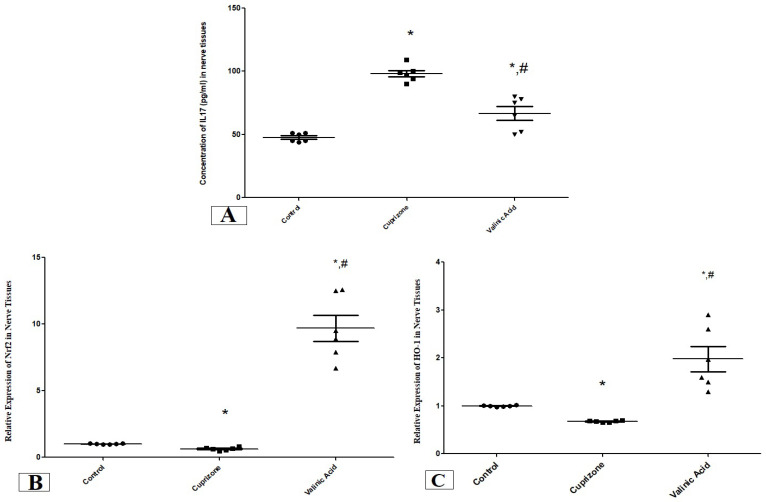
The concentration of IL17 (pg/mL) (**A**) and the expression of antioxidant genes (Nrf2, (**B**) and HO-1, (**C**)) in the sciatic nerve in different groups. * Significant vs. control group, # significant vs. Cuprizone group. *p* < 0.05.

**Figure 3 brainsci-14-00012-f003:**
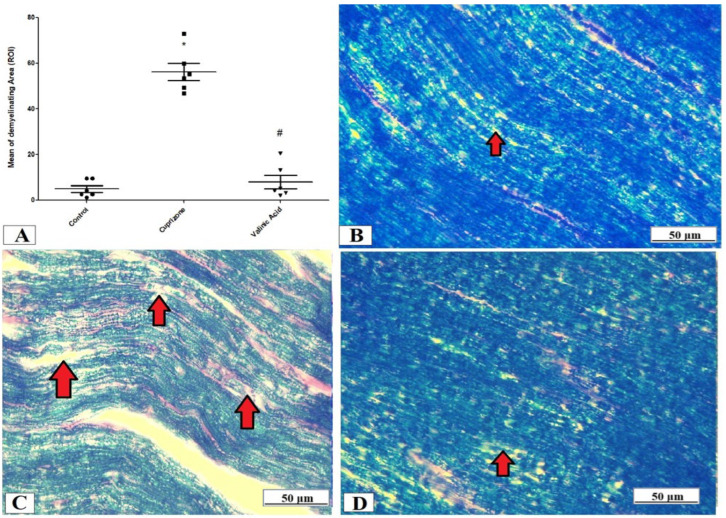
Luxol fast blue (LFB) staining for myelination of the sciatic nerve. (**A**) the score of demyelinating areas in different studied groups. Photomicrographs of LFB staining from the control group.(**B**), Cuprizone group (**C**), and VA group (**D**). Red arrows indicate the areas of demyelination. * Significant vs. control group, # significant vs. Cuprizone group. *p* < 0.05.

**Figure 4 brainsci-14-00012-f004:**
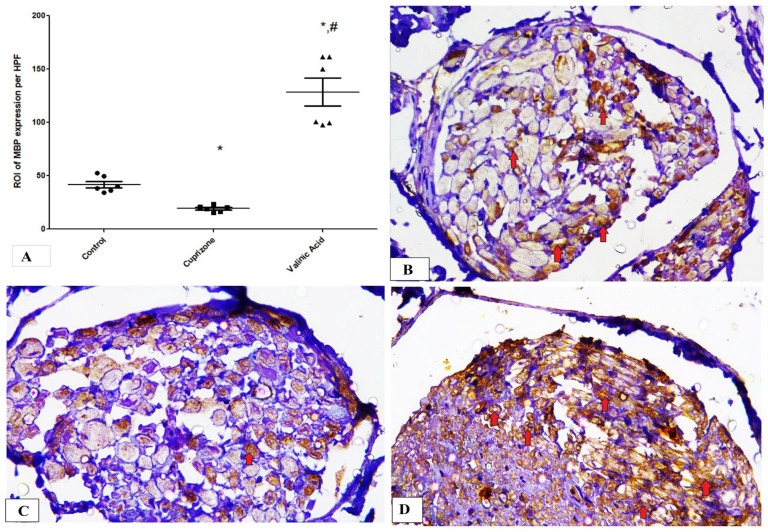
Immunohistopathological staining for myelin basic protein (MBP) in sciatic nerve. (**A**) the score of MBP expression in different studied groups. Photomicrographs of MBP staining from control group.(**B**), Cuprizone group (**C**), and VA group (**D**). Red arrows indicate brown staining for myelin basic protein (MBP) * significant vs. control group, # significant vs. Cuprizone group. *p* < 0.05.

**Figure 5 brainsci-14-00012-f005:**
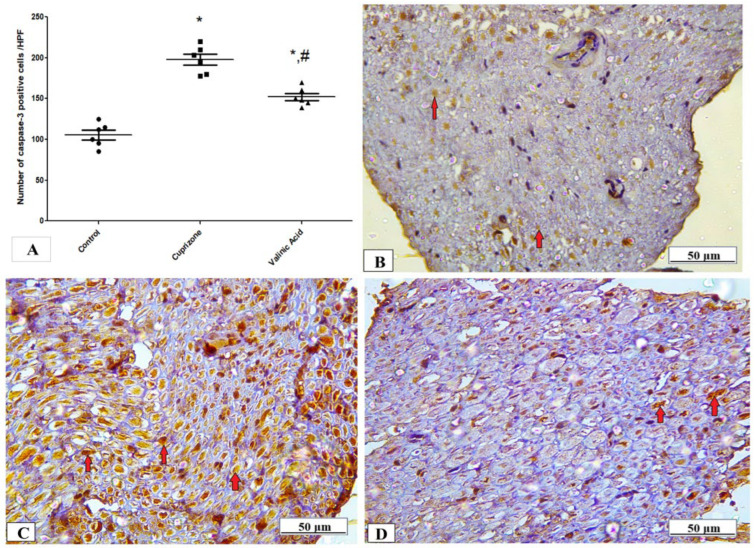
Immunohistopathological staining for caspase-3 in sciatic nerve. (**A**) the score of caspase-3 expression in different studied groups. Photomicrographs of caspase-3 from the control group.(**B**), Cuprizone group (**C**), and VA group (**D**). * Significant vs. control group, # significant vs. Cuprizone group. *p* < 0.05.

**Figure 6 brainsci-14-00012-f006:**
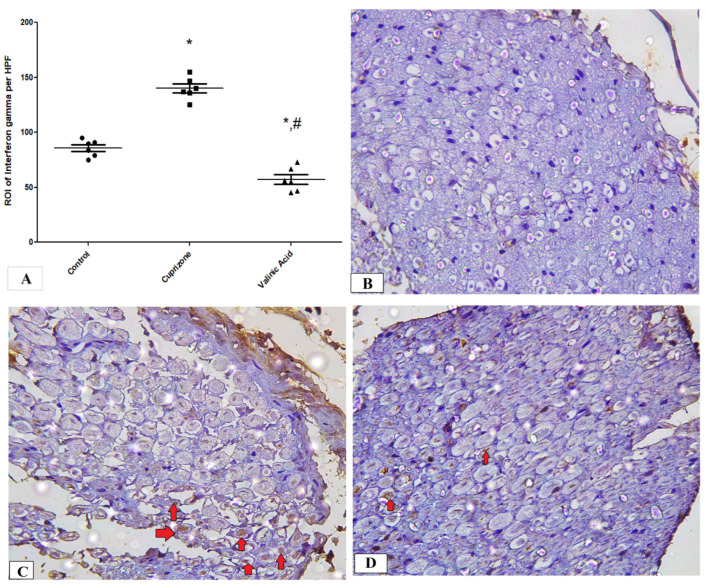
Immunohistopathological staining for INF-γ in sciatic nerve. (**A**) the score of INF-γ expression in different studied groups. Photomicrographs of INF-γ from the control group. (**B**), Cuprizone group (**C**), and VA group (**D**). red arrows indicate brown staining for INF-γ * significant vs. control group, # significant vs. Cuprizone group. * *p* < 0.05.

**Figure 7 brainsci-14-00012-f007:**
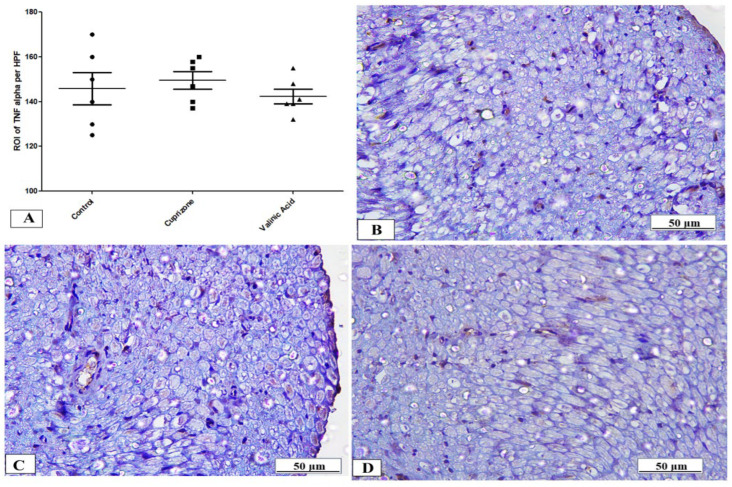
Immunohistopathological staining for TNF-α in sciatic nerve. (**A**) the score of TNF-α expression in different studied groups. Photomicrographs of TNF-α from the control group (**B**), Cuprizone group (**C**), and VA group (**D**).

**Figure 8 brainsci-14-00012-f008:**
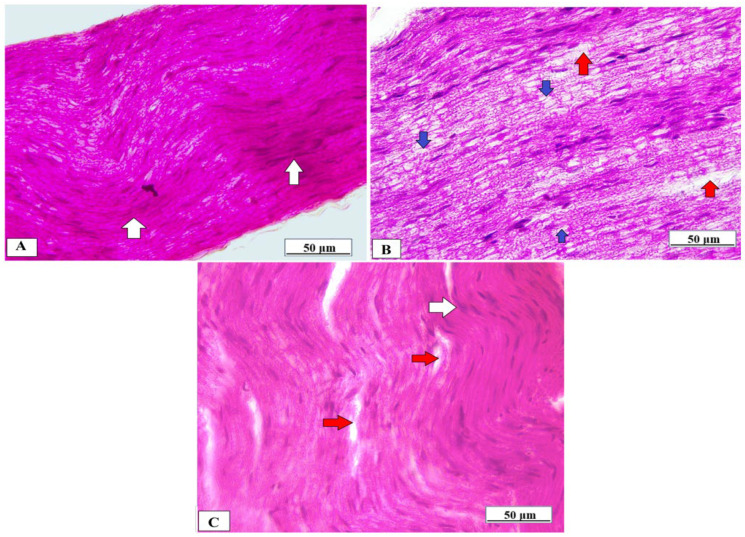
Histopathological examination of the sciatic nerve using H&E from different studied groups. (**A**) photomicrograph from the normal control group showing dense, uniform, regular myelinated nerve fibers arranged in concentric layers with a regular arrangement of Schwann cell nuclei (white arrows). (**B**) = photomicrograph from the Cuprizone group showing that myelinated fibers are broad, loose, and partially demyelinated (red arrows) with an irregular pattern of clear vacuoles, i.e., bubbling (blue arrows) and (**C**) = photomicrograph from vanillic acid-treated group revealed regular myelin sheaths (white arrows).

**Figure 9 brainsci-14-00012-f009:**
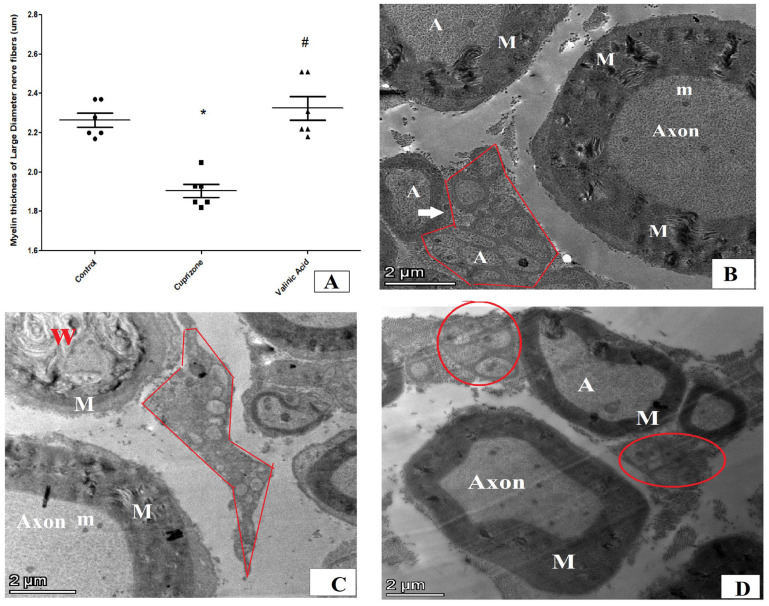
Ultra-microscopic structure of the sciatic nerve via electron microscopy for different studied groups. (**A**) = score of myelin sheaths’ thickness in micrometers in different groups. (**B**) = photomicrograph from normal control group of rats showing large myelinated nerve fibers with thick myelin sheath (M) and its axonal (A) cytoplasm containing mitochondria (m) with small-diameter nerve fiber axons surrounded by Schwann cells ensheathing each axon in a pocket of its cytoplasm, forming a Remak bundle (red circle). (**C**) = photomicrograph from the Cuprizone group showing endoneurial edema (manifested by the distinct displacement between nerve fibers), Wallerian degeneration (W) in axons of large-diameter fibers and degeneration of non-myelinated nerve fibers surrounded by Schwan cells. (**D**) = photomicrograph from vanillic acid-treated group showing intact large- and small-diameter axons near the normal control group. * significant vs. control group, # significant vs. Cuprizone group. *p* < 0.05.

## Data Availability

The raw data of the current study were available on request. The data are not publicly available due to privacy.

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
