# Peer review of "Vanillic Acid Ameliorates Demyelination in a Cuprizone-Induced Multiple Sclerosis Rat Model: Possible Underlying Mechanisms"

_brainsci, 2023, doi:10.3390/brainsci14010012_

Round 1

Reviewer 1 Report

Comments and Suggestions for Authors

The original article entitled “Vanillic acid Ameliorates Demyelination in Cuprizone-induced Multiple Sclerosis Rat Model: Possible Underlying Mechanisms” written by Sally M. Safwat et al., is very interesting for readers.

       The topic of the present manuscript is relevant on the field;

- the introduction provides sufficient background and includes relevant references;

- the design research is well explained,

- the conclusions are consistent because evidence the presented arguments, but more research is needed in the future studies;

However, this article cannot be published in the present form because of some questions, which it arises.

1/Please change the format of figures

2/Please add these references to explain the cross talk between astrocyte and oligodendrocytes (doi: 10.1016/j.bbrc.2023.02.066)

Comments on the Quality of English Language

Extensive editing of English language required

Author Response

Reviewer’s # 1

Comment

  • The original article entitled “Vanillic acid Ameliorates Demyelination in Cuprizone-induced Multiple Sclerosis Rat Model: Possible Underlying Mechanisms” written by Sally M. Safwat et al., is very interesting for readers.
  • The topic of the present manuscript is relevant on the field;
    • the introduction provides sufficient background and includes relevant references;
    • the design research is well explained,
    • the conclusions are consistent because evidence the presented arguments, but more research is needed in the future studies;

Response

  • Great thanks

However, this article cannot be published in the present form because of some questions, which it arises.

Comment

1/Please change the format of figures

Response

Done

Comment

2/Please add these references to explain the cross talk between astrocyte and oligodendrocytes (doi: 10.1016/j.bbrc.2023.02.066)

Response

  • The following statement was added to the discussion “Recently, ELBini and Neili, (2023) demonstrated that the use of K channel blocker stimulated the process of myelination of nerve fibers in a model of multiple sclerosis when introduced early during disease development or during remission and exacerbated the demyelination when introduced in toxic inflammatory environment during the inflammatory processes”
  • This reference was added to the reference list
  • ELBini, I. and Neili, N.E., 2023. Potassium channels at the crossroads of neuroinflammation and myelination in experimental models of multiple sclerosis. Biochemical and Biophysical Research Communications653, pp.140-146.

Comment

Extensive editing of English language required

Response

English editing was done in the revised manuscript with tracking for these changes in the edited manuscript

Reviewer 2 Report

Comments and Suggestions for Authors

In the paper entitled “Vanillic acid Ameliorates Demyelination in Cuprizone-induced Multiple Sclerosis Rat Model: Possible Underlying Mechanisms” Safwat and co-workers investigated the effect of the natural compound vanillic acid on demyelination/remyelination dynamics in isolated sciatic nerve from cuprizone-treated mice. They also evaluated changes of genes related to antioxidant and pro-inflammatory functions.

The experimental plan is appropriate, but the poor quality of the pictures may alter the results. 

Introduction:

- The terminology Progressive-relapsing MS is no more uses. These patients are now classified as primary progressive MS (active or not active)

- Check the sentence: “(Lassmann & Bradl, 2017) were established for the proper under-standing of its pathophysiological mechanisms”.

-  The sentence “Natural substances are almost often the first choice for treating various disorders” is too strong.

Materials and Methods:

- IHC protocol and analysis (densitometric quantifications of LFB, MBP and cytokines) should be explained more in detail (e.g., incubation time, Ab dilution)

- Details about the kits used for RNA extraction and real time PCR are missing.

Results:

- Data reported in Table 1 should be visualized by means of histograms.

- The quality of the pictures used for all the densitometric analysis is very poor, not suitable for publication. The counterstain prevails on the specific signal. The LFB signal is overexposed. I do not consider these images analysable, so I suggest repeating the immunohistochemical staining.

Comments on the Quality of English Language

English language editing/proofreading would significantly benefit the manuscript. The meaning intended by the authors is generally understandable, but the quality of the writing and grammar in the text is patchy and often poor.

Author Response

Reviewer’s 2

In the paper entitled “Vanillic acid Ameliorates Demyelination in Cuprizone-induced Multiple Sclerosis Rat Model: Possible Underlying Mechanisms” Safwat and co-workers investigated the effect of the natural compound vanillic acid on demyelination/remyelination dynamics in isolated sciatic nerve from cuprizone-treated mice. They also evaluated changes of genes related to antioxidant and pro-inflammatory functions.

The experimental plan is appropriate, but the poor quality of the pictures may alter the results. 

Comment

Introduction:

- The terminology Progressive-relapsing MS is no more uses. These patients are now classified as primary progressive MS (active or not active)

Response

Great thanks. This point was corrected in revised manuscript

Comment

- Check the sentence: “(Lassmann & Bradl, 2017) were established for the proper under-standing of its pathophysiological mechanisms”.

Response

Great thanks. This typing error was corrected

Comment

-  The sentence “Natural substances are almost often the first choice for treating various disorders” is too strong.

Response

This sentence was corrected.

Materials and Methods:

Comment

- IHC protocol and analysis (densitometric quantifications of LFB, MBP and cytokines) should be explained more in detail (e.g., incubation time, Ab dilution)

Response

We mentioned the reference for each technique to avoid pilgarism. We added brief informations about the dilution for each antibody and its incubation times.

Comment

- Details about the kits used for RNA extraction and real time PCR are missing.

Response

Done

Results:

Comment

- Data reported in Table 1 should be visualized by means of histograms.

Response

Done

Comment

- The quality of the pictures used for all the densitometric analysis is very poor, not suitable for publication. The counterstain prevails on the specific signal. The LFB signal is overexposed. I do not consider these images analysable, so I suggest repeating the immunohistochemical staining.

Response

We repeated immunostaining and new figures were added

Comment

Comments on the Quality of English Language

English language editing/proofreading would significantly benefit the manuscript. The meaning intended by the authors is generally understandable, but the quality of the writing and grammar in the text is patchy and often poor.

Response

English editing was done in the revised manuscript with tracking for these changes in the edited manuscript

Reviewer 3 Report

Comments and Suggestions for Authors

Review

The authors presented the study to explore the effects of Vanillic acid (VA) on cuprizone (Cup) demyelinating rat model and the mechanisms behind such effects. The method is straightforward and the figures are easy to follow. This paper addressed a significant clinical treatment. My major concern in this paper is its poor data presentation, confusing formatting, and weak discussion to support the conclusion.

Following are my major and minor comment points.

Major points:

1.     In methods 2.1, why do you include male rats only? Sex equity in animal studies should be considered.

2.     For all bar graphs, please show dot plots so we can visualize individual data points.

3.     For figure 2A, I suggest changing to quantification of myelination area. Because Figure 2B-D demonstrates the opposite trends, which could be misleading.

4.     Figure 4B, please make sure this is the correct magnification. Figure 4B looks much different from C and D at cellular level.

5.     IFN-gamma is lower in vanillic acid group when compared to normal group. Please explain.

Comments on the Quality of English Language

1.     Language, typos and grammar should be carefully edited and corrected. For example:

a.     Abstract: “The nerve conduc-tion study (NCV) in isolated sciatic nerve” is. Incomplete

b.     Figure 5: IFN-gamma, not INF

Author Response

Reviewer’s # 3

The authors presented the study to explore the effects of Vanillic acid (VA) on cuprizone (Cup) demyelinating rat model and the mechanisms behind such effects. The method is straightforward and the figures are easy to follow. This paper addressed a significant clinical treatment. My major concern in this paper is its poor data presentation, confusing formatting, and weak discussion to support the conclusion.

Following are my major and minor comment points.

Major points:

Comment

  1. In methods 2.1, why do you include male rats only? Sex equity in animal studies should be considered.

Response

  • Male rats were chosen in this study on basis of previous study by Ünsal, C., & Özcan, M. (2018), who concluded that cuprizone neurotoxicity affects both genders of rats, but the effect is typically more prominent in males and cuprizone-induced demyelination may have a good potential for investigating the peripheral MS-related effects.
  • Ünsal, C., & Özcan, M. (2018). Neurotoxicity of cuprizone in female and male rats: Electrophysiological observations. Neurophysiology50, 108-115.‏
  • This sentence was added to revised manuscript

Comment

  1. For all bar graphs, please show dot plots so we can visualize individual data points.

Response

Done

Comment

  1. For figure 2A, I suggest changing to quantification of myelination area. Because Figure 2B-D demonstrates the opposite trends, which could be misleading.

Response

We did staining for LFB stain again and new image was added to the revised manuscript

Comment

  1. Figure 4B, please make sure this is the correct magnification. Figure 4B looks much different from C and D at cellular level.

Response

We added new figure containing images of the same magnifications

Comment

  1. IFN-gamma is lower in vanillic acid group when compared to normal group. Please explain.

Response

We did not have any explanation according to our results but this was just a finding and we added this point to the discussion section

Comment

Comments on the Quality of English Language

  1. Language, typos and grammar should be carefully edited and corrected. For example:
  2. Abstract: “The nerve conduction study (NCV) in isolated sciatic nerve” is. Incomplete

Response

Done

Comment

  1. Figure 5: IFN-gamma, not INF

Response

done

Comment

Quality of English editing

Response

English editing was done in the revised manuscript with tracking for these changes in the edited manuscript

Round 2

Reviewer 1 Report

Comments and Suggestions for Authors

I accept the article in this form

Comments on the Quality of English Language

Minor editing of English language required

Reviewer 2 Report

Comments and Suggestions for Authors

The quality of the paper is now acceptable for its publication in Brain Sciences.

Reviewer 3 Report

Comments and Suggestions for Authors

The authors have addressed all my concerns.